

# Differential responses of the acidobacterial community in the topsoil and subsoil to fire disturbance in *Pinus tabulaeformis* stands

Weike Li[1,2], Xiaodong Liu[1] and Shukui Niu[1]

[1] Beijing Key Laboratory for Forest Resources and Ecosystem Processes, Beijing Forestry University, Beijing, China

[2] Fire Engineering, China Fire and Rescue Institute, Beijing, China

## ABSTRACT

*Acidobacteria* is found to be dominant and abundant in forest soil, and performs specific ecological functions (such as cellulose decomposition and photosynthetic capacity, etc.). However, relative limited is known about its changing patterns after a fire interruption. In this study, the response of soil *Acidobacteria* to a wildfire disturbance was investigated using the Illumina MiSeq sequencing system. The research area was classified by different severities of fire damage (high, moderate, and low severity, and an unburnt area), and samples were collected from various soil layers (0–10 cm as topsoil; 10–20 cm as subsoil). We obtained a total of 986,036 sequence reads; 31.77% of them belonged to *Acidobacteria*. Overall, 18 different *Acidobacteria* subgroups were detected, with subgroups 4, 6, 1, 3, and 2 the most abundant, accounting for 31.55%, 30.84%, 17.42%, 6.02%, and 5.81% of acidobacterial sequences across all samples, respectively. Although no significant differences in acidobacterial diversity were found in the same soil layer across different fire severities, we observed significantly lower numbers of reads, but higher Shannon and Simpson indices, in the topsoil of the high-severity fire area than in the subsoil. Non-metric multidimensional scaling (NMDS) analysis and permutational multivariate analysis of variance (PERMANOVA) also revealed significant differences in the acidobacterial community structure between the two soil layers. Soil pH, total nitrogen, $NH_4^+$-N, the Shannon index of understory vegetation and canopy density were the major drivers for acidobacterial community structure in the topsoil, while soil pH and organic matter were significant factors in the subsoil. A variance partitioning analysis (VPA) showed that edaphic factors explained the highest variation both in the topsoil (15.6%) and subsoil (56.3%). However, there are large gaps in the understanding of this field of research that still need to be explored in future studies.

## INTRODUCTION

Fire is a significant interfering factor in forest ecosystems. Heat from fire has strong direct and indirect effects on vegetation (*Knelman et al., 2015*), animals

Corresponding authors
Xiaodong Liu, xd_liu@bjfu.edu.cn
Shukui Niu, niushukui@bjfu.edu.cn

(*Wang & Yan, 2017*), soil (*Mikita-Barbato, Kelly & Tate, 2015*), regional climate (*Lehmann et al., 2014*), and the water cycle (*Chen et al., 2013*). As an important part of the forest ecosystem, soil microorganisms maintain ecosystem stability through the role in driving organic matter decomposition and soil nutrient cycling (*Kristal & Kathleenk, 2010*). Although the response of soil microorganisms to fire disturbance has been studied extensively, the impacts of fire severity on bacterial communities is far from a settled question. *Pressler, Moore & Cotrufo (2018)* concluded that there are insufficient studies to draw strong conclusions about the role of fire severity on the bacterial community. *Weber et al. (2014)* found inconsistent response of bacterial communities to fire severity. However, the above studies focused on the overall changes in bacterial communities. Little attention has been paid to the effects of a wildfire on specific bacterial phyla.

*Acidobacteria* have been recognized as being ubiquitous and abundant in forest soil (*Jones et al., 2009*; *Mbuthia et al., 2015*). Based on 16S rRNA sequencing, 10% to 50% of the obtained soil microbial sequences have been assigned to this phylum (*Navarrete et al., 2013*; *Liu et al., 2016*). Currently, *Acidobacteria* are classified into 26 subgroups (*Barns et al., 2007*); subgroups 1 to 4 and subgroup 6 are found to be predominant in soils (*Naether et al., 2012*; *Zhang et al., 2014*). Although bacteria of this phylum are difficult to cultivate (*Liu et al., 2016*), molecular techniques provide an opportunity to gain an insight into their ecological and physiological characteristics (*Kielak et al., 2016*). In previous studies, *Acidobacteria* have been observed to exhibit oligotrophy that is found to be related to some ecosystem processes, such as the decomposition of organic matter and carbon cycling (*Jones et al., 2009*; *Griffiths et al., 2011*; *Eichorst, Kuske & Schmidt, 2011*; *Sul et al., 2013*) and their phylogenetic diversity is considered to be comparable to that of the phylum *Proteobacteria* (*Liu et al., 2016*). However, not all members of the subgroups show consistent biological characteristics. For example, the gene sequences of *Acidobacteria* are also detected in neutral or even alkaline environments (*Xiong et al., 2012*). In addition, *De Castro et al. (2013)* reported that several isolates of this phylum are capable of growth in a eutrophic environment. Furthermore, *Belova et al. (2018)* reported two acidobacterial strains (SBC82[T] and CCO287) were assigned to a novel genus and species and members of this genus colonize acidic soils and peatlands and specialize in degrading complex polysaccharides. *Hausmann et al. (2018)* showed by genome-centric metagenomics and metatranscriptomics that members of the *Acidobacteria* phylum have a putative role in peatland sulfur cycling. *Eichorst, Kuske & Schmidt (2011)* performed a large-scale comparative genome analysis of acidobacterial subdivisions and observed that low- and high-affinity respiratory oxygen reductases in multiple genomes, suggesting the capacity for growing across different oxygen gradients. They also detected the capacity of many acidobacterial genomes to use multiple carbohydrates as well as inorganic and organic nitrogen sources. *Diamond et al. (2019)* also reported that acidobacteria play important roles in C cycling in soils.

Studies of the influence of fire on the abundance of *Acidobacteria* have shown inconsistent results, with reports of a great decrease (*Xiang et al., 2014*), a significant increase in lightly burnt unvegetated soils (*Knelman et al., 2015*), and minor changes at the phylum level, but with a significantly lower abundance of subgroup 1 after burning (*Shen, Chen & Lewis,*

*2016*). The differences in these results may be attributable to the sampling time, location, fire severity, vegetation type, etc., as well as the variable biological characteristics of the *Acidobacteria* subgroups (*Liu et al., 2016*). For example, *Zhang et al. (2014)* reported that the relative abundances of subgroup 1, 2, 3 increased with the altitude, but the relative abundances of subgroup 4, 5, 6, 16, 17 decreased with the altitude. Several researchers also found that the effects of soil pH on *Acidobacteria* subgroups are not exactly the same (*Jones et al., 2009*; *Turlapati et al., 2013*; *Kielak et al., 2016*). *Xiang et al. (2014)* reported that the relative abundance of *Acidobacteria* is negatively correlated with soil pH; however, *Liu et al. (2016)* found a significantly positive relationship between the abundance of *Acidobacteria* and soil pH. Although significant findings have been made concerning the distribution of *Acidobacteria* after fire disturbance (*Xiang et al., 2014*; *Shen, Chen & Lewis, 2016*; *Sun et al., 2016*), knowledge of the response pattern at the subgroup level is still limited. In this study, we try to control other variables to be unified and mainly to explore the influence of different fire severities on *Acidobacteria*.

In the context of the high relative abundance and special ecological characteristics of *Acidobacteria*, we aimed to gain an insight into the response of this phylum (especially at the subgroup level) to wildfire. We assessed the response of *Acidobacteria* to different fire severities (high, moderate and low severity, and unburnt) at two soil depths (0–10 cm as the topsoil and 10–20 cm as the subsoil). We sought to address three questions: (1) How does the acidobacterial community composition vary in the topsoil and subsoil after burning at different severities? (2) What are the main factors that affect changes in soil acidobacterial communities after a wildfire? (3) Are the driving factors in the topsoil and subsoil the same?

## MATERIALS & METHODS

### Site description and soil sampling

This study was conducted in Pingquan County, Hebei province, northern China (118°22′∼118°37′E, 41°01′∼41°21′N) (Fig. S1). As a typical coniferous tree species in North China, *Pinus tabulaeformis* is widely distributed in this area (*Zhu et al., 2015*). The oil content of the branches and leaves of *Pinus tabulaeformis* is high, and the trees belong to the flammable forest type (*Niu, 2011*). The soil types were classified as brown soil and cinnamon soil (*Li et al., 2015*). The climate is semi-humid and semi-humid continental monsoon, with a mean annual temperature of 7.3 °C. The mean annual precipitation in this region is 550 mm, of which 70% is summer rainfall (July–September) (*Li et al., 2015*).

In April 2015, a 56.33 ha wildfire occurred in this region. The burned area was divided into different regions according to the fire severity (high, moderate and low severity, and a nearby unburnt area) (*Zheng, Cui & Di, 2012*). The percentages of different severity levels are about 55.56% (high), 33.33% (moderate), 11.11% (low). The classification standard of fire severity is shown in Table 1 and we also provide a set of photographs to show the visual perception of different fire severity areas (Fig. 1). Soil samples were collected 6 months after the fire. Three plots (20 m x 20 m) were established in each of four distinct patches, representing the different fire intensity levels (high, moderate, low and unburned).

**Table 1  Criteria for the classification of forest fire severity (*Zheng, Cui & Di, 2012*).**

| Fire severity | Criteria for the classification |
|---|---|
| High severity | Trees with more than 80% burns or burned to death, the undergrowth shrubs all consumed, char height of trunks were more than 5 m. As a result of the ground organic matter all burned, mineral soil color and structure are changed. |
| Moderate severity | 10% to 80% of the trees were burned and the char height of trunks was between 2 to 5 m. The upper part of forest litter was burned. |
| Low severity | Trees were burned less than 10%, part of the undergrowth shrubs was consumed (below 40%), char height of trunks were below 2 m. |
| Control | Unburned |

There were 12 plots in total, with similar slopes ($20\sim23°$), aspects ($37\sim42°$) and elevations ($1{,}119\sim1{,}143$ m). In each plot, five random soil cores at a depth of 0–10 cm and 10–20 cm were taken. Before sample collection, the litter, charred debris, and ash layer were removed. Then, the five soil cores from the same layer were mixed into one soil sample. In total, 24 composite samples (12 topsoil; 12 subsoil) were collected. All samples were transported to the laboratory in an ice box for further analysis. Samples were divided into two parts: one part was stored at 4 °C for biogeochemical analysis, and the other was stored at −20 °C for DNA analysis.

## Plant diversity and structure and soil physicochemical analyses

The plant species, plant height, canopy density and tree diameter at breast height (DBH > 5 cm) were recorded in each plot (Table S1). Then, ten quadrats were randomly established in each plot (five $1 \times 1$ m quadrants for herbs; five $5 \times 5$ m quadrants for shrubs). The diversity of the understory vegetation (i.e., herbs and shrubs) was evaluated using the Shannon-Wiener index ($H'_{Vegetation}$), Simpson index ($D'_{Vegetation}$) and Pielou's evenness index ($J'_{Vegetation}$) (*Yu et al., 2012*).

Soil moisture (SM) was determined by oven-drying the samples at 105 °C until they were of constant weight. pH was determined using a pH meter for a 1:2.5 ratio of fresh soil to deionized water at 20 °C. Soil organic matter (OM) content was measured by dichromate oxidation (*Xue, Li & Chen, 2014*), and total nitrogen (TN) was measured with the Kjeldahl method (*Walkley & Black, 1934*). The ammonium nitrogen ($NH_4^+$-N) and nitrate nitrogen ($NO_3^-$-N) contents were also measured (*Crooke & Simpson, 1971*; *Best, 1976*).

## DNA extraction and PCR amplification

Genomic DNA was extracted from 0.5 g of fresh soil using an E.Z.N.A.® soil DNA kit (Omega Bio-tek, Norcross, GA, USA), following the manufacturer's instructions. The quality of the DNA extracted was checked by 1% agarose gel electrophoresis and spectrophotometry (the ratio of optical density at 260 nm/280 nm). All extracted DNA samples were stored at −20 °C for further analysis.

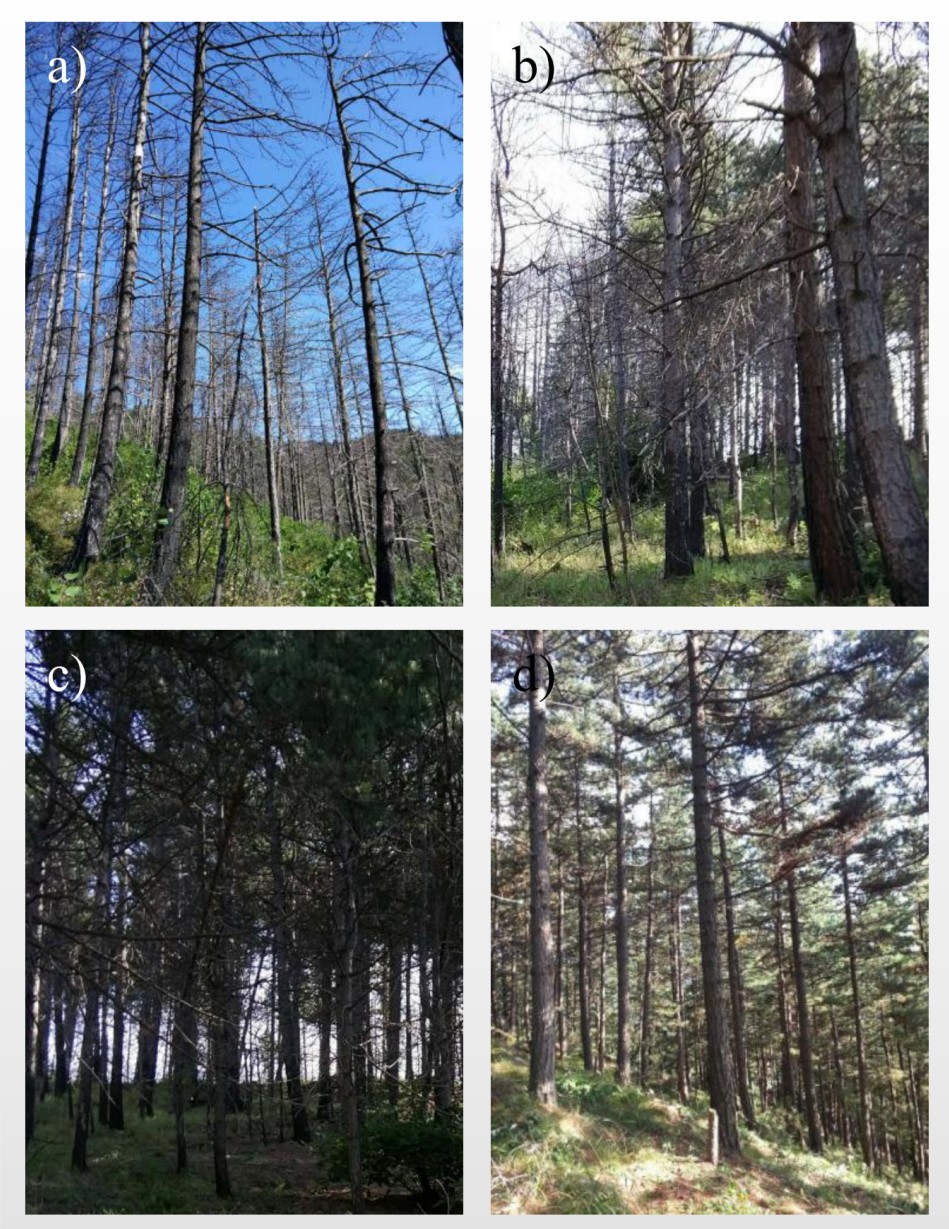

**Figure 1** **Photographs depicting different fire severity areas.** (A) high severity area (B) moderate severity area (C) low severity area (D) unburnt control area. Photographs were taken in October 2015.

To assess the bacterial composition, the V3–V4 hypervariable regions of the bacterial 16S rRNA genes were amplified by PCR (95 °C for 5 min; 33 cycles at 95 °C for 30 s, 56 °C for 30 s and 72 °C for 40 s; with a final extension of 72 °C for 10 min), using the universal primers forward 338F (5′-ACTCCTACGGGAGGCAGCAG-3′) and reverse 806R (5′-GGACTACHVGGGTWTCTAAT-3′) (*Liu et al., 2018*). These primers contained an 8 nucleotide barcode sequence unique to each sample. PCR reactions were performed in

triplicate in a 50 μL mixture containing 5 μL of 10× Pyrobest Buffer, 4 μL of 2.5 mM dNTPs, 2 μL of each primer (10 μM), 0.3 μL of Pyrobest DNA Polymerase (2.5 U/ μL; TaKaRa code DR005A) and 30 ng of template DNA. Then, the mixture was used to perform sequencing. The raw reads were deposited into the National Center for Biotechnology Information Sequence Read Archive database (accession number: SRP158101).

## Illumina MiSeq sequencing

Amplicons were extracted from 2% agarose gels and purified using an AxyPrep DNA Gel Extraction kit (Axygen Biosciences, Union City, CA, USA) according to the manufacturer's instructions, and quantified using QuantiFluor-ST (Promega, Madison City, WI, USA). Purified amplicons were pooled in equimolar amounts and paired-end sequenced (2×300) on an Illumina MiSeq platform (Beijing Allwegene Technology Co., Ltd, China) according to standard protocols.

## Processing of sequencing data

The extraction of high-quality sequences was performed with the QIIME package (Quantitative Insights Into Microbial Ecology) (version 1.2.1) (http://qiime.org/). Raw sequences were selected on the basis of sequence length, quality, primer, and tag. The raw sequences were selected and the low-quality sequences were removed as follows: (1) 300 bp reads were truncated at any site receiving a mean quality score of <20 over a 10 bp sliding window, truncated reads that were shorter than 50 bp were discarded; (2) exact barcode matching, 2 nucleotide mismatch in primer matching, reads containing ambiguous characters were removed; (3) only sequences with an overlap longer than 10 bp were assembled according to their overlap sequence. Reads that could not be assembled were discarded. Chimeric sequences were identified and removed based on the UCHIME algorithm (*Edgar et al., 2011*). The unique sequence set was classified into operational taxonomic units (OTUs) under the threshold of 97% identity using UCLUST (*Edgar, 2010*) and then the taxonomic annotations were conducted for each OTU based on the Silva119 16S rRNA database using a confidence threshold of 90% (*Quast et al., 2013*). OTUs with less than five reads were removed to reduce the risk of artificially inflating richness due to sequencing errors (*Chen et al., 2017*).

## Data analysis

We performed separate one-way analysis of variance (ANOVA) with Tukey's test (equal variances) or Dunnett's T3 test (unequal variances) to determine the effects of wildfire of varying severity on soil physicochemical properties and the composition and diversity of the acidobacterial community in different soil layers (0–10 and 10–20 cm). The differences in the composition and diversity of the acidobacterial community between the topsoil and subsoil within different fire severities were also compared by independent-sample $t$-test. Pearson correlation analysis was used to determine relationships between the abundance of *Acidobacteria* subgroups and environmental variables. Differences were considered significant with a $P$ value of 0.05 as the threshold. All tests were carried out using SPSS 19.0 for Windows.

**Table 2  Soil physical and chemical properties.**

| Soil properties | Soil layers | Fire severity | | | | One-way ANOVA | |
|---|---|---|---|---|---|---|---|
| | | High severity | Moderate severity | Low severity | Unburned control | $f$-values | $p$-values |
| OM ($mg\ kg^{-1}$) | topsoil | $9,190 \pm 90$b | $17,130 \pm 760$a | $17,460 \pm 1230$a | $16,160 \pm 550$a | 76.11 | 0.000 |
| | subsoil | $4,000 \pm 490$b | $6,910 \pm 590$a | $7,390 \pm 1,180$a | $7,920 \pm 440$a | 16.89 | 0.001 |
| TN ($mg\ kg^{-1}$) | topsoil | $310 \pm 10$c | $560 \pm 40$b | $850 \pm 30$a | $880 \pm 50$a | 171.71 | 0.000 |
| | subsoil | $180 \pm 10$d | $270 \pm 40$c | $400 \pm 10$b | $660 \pm 40$a | 169.47 | 0.000 |
| $NH_4^+$-N ($mg\ kg^{-1}$) | topsoil | $1.72 \pm 0.03$c | $1.45 \pm 0.37$c | $3.70 \pm 0.62$b | $19.49 \pm 0.90$a | 674.98 | 0.000 |
| | subsoil | $1.25 \pm 0.30$c | $3.01 \pm 0.68$c | $11.78 \pm 0.67$b | $16.32 \pm 1.07$a | 285.64 | 0.000 |
| $NO_3^-$-N ($mg\ kg^{-1}$) | topsoil | $1.64 \pm 0.43$b | $2.45 \pm 0.84$a | $3.56 \pm 0.36$a | $2.64 \pm 0.06$a | 7.29 | 0.011 |
| | subsoil | $2.10 \pm 0.62$b | $2.10 \pm 0.44$b | $3.87 \pm 0.72$a | $3.16 \pm 0.54$a | 6.49 | 0.015 |
| SM (%) | topsoil | $17.23 \pm 1.20$a | $10.22 \pm 1.44$c | $11.04 \pm 1.31$c | $13.43 \pm 0.68$b | 20.72 | 0.000 |
| | subsoil | $11.39 \pm 0.10$b | $7.21 \pm 0.68$c | $11.25 \pm 0.99$b | $15.41 \pm 0.66$a | 71.59 | 0.000 |
| pH | topsoil | $6.90 \pm 0.27$a | $6.13 \pm 0.12$b | $5.68 \pm 0.38$b | $5.65 \pm 0.25$b | 13.89 | 0.002 |
| | subsoil | $6.19 \pm 0.03$a | $5.67 \pm 0.14$a | $5.62 \pm 0.49$a | $5.61 \pm 0.44$a | 2.07 | 0.183 |

**Notes.**
OM, organic matter; TN, total nitrogen; $NH_4^+$-N, , ammonium nitrogen; $NO_4^-$-N, nitrate nitrogen; SM, soil moisture.
Different lowercase letters after the entries indicate significant differences at the same soil depth of different sample plots at $P < 0.05$. Values are means $\pm$ standard deviations ($n = 3$).

To correct for sampling efficiency, we used a randomly selected subset of 6,199 acidobacterial sequences (the minimum sample size among 24 samples) per sample for downstream analysis. Shannon–Wiener ($H'_{Acidobacteria}$) and Simpson ($D'_{Acidobacteria}$) indices of the acidobacterial community were calculated using the OTUs with 97% identity. Non-metric multidimensional scaling (NMDS) analysis based on Bray-Curtis and Weighted-Unifrac distance and permutational multivariate analysis of variance (PERMANOVA) were used to compare the composition of the acidobacterial community between different fire-damaged areas. Redundancy analysis (RDA) was carried out to assess the relationships between soil properties and acidobacterial community data (relative abundance of OTUs), and only environmental factors with variance inflation factor values of less than 20 were selected for the model (*Xiang et al., 2014*). A variance partitioning analysis (VPA) was further performed to quantify the effects of environmental factors on acidobacterial community structure. The above analyses were carried out with the Vegan package (*Jari et al., 2018*) in R (*R Core Team, 2017*).

## RESULTS

### Soil properties and understory vegetation

Table 2 presents basic information on the physical and chemical properties of the soil after burning. One-way ANOVA showed that soil OM, TN, ammonium nitrogen ($NH_4^+$-N), and nitrate nitrogen ($NO_3^-$-N) in both soil layers significantly decreased with an increase in fire severity. However, soil pH showed the opposite trend in that it increased after burning. Interestingly, the patterns of change of SM in the topsoil and subsoil were inconsistent. The highest value of SM was observed in the topsoil of the high-severity fire area.

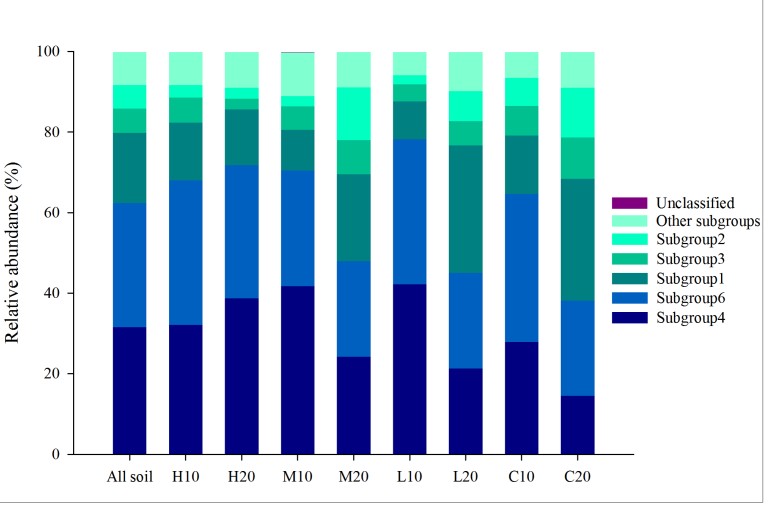

**Figure 2** **Relative abundance of dominant acidobacterial subgroups in areas affected by differing fire severity.** H, high severity; M, moderate severity; L, low severity; C, unburnt control; 10, topsoil (0–10 cm); 20, subsoil (10–20 cm).

As shown in Fig. S2 and Table S2, the diversity of the understory vegetation was significantly changed after a high-severity wildfire. The highest value of Shannon-Wiener (H'$_{Vegetation}$) and Simpson (D'$_{Vegetation}$) indices and the lowest value of Pielou's evenness index (J'$_{Vegetation}$) were observed in the high-severity area compared with the other three types of plots ($P < 0.05$). However, no significant differences were found among the moderate-severity, low-severity, and unburnt areas.

### *Acidobacteria* community composition and structure

A total of 986,036 reads were detected using 24 samples through Illumina MiSeq sequencing. Of these reads, *Acidobacteria* is the most abundant phylum with a abundance of 31.77% (Fig. S3). Each sample contained 6,199 to 27,585 acidobacterial reads (mean = 13,053), with different phylotypes (OTUs) ranging from 173 to 370 (mean = 309; total = 7, 425). The *Acidobacteria* sequences obtained were further classified into 18 subgroups. Among them, subgroups 4, 6, 1, 3 and 2 were found to be dominant (abundance >5%) and common to the 24 libraries, which accounted for 31.55%, 30.84%, 17.42%, 6.02% and 5.81% of the total acidobacterial sequences across all soils, respectively (Fig. 2). The other 13 subgroups (subgroups 5, 7, 10–13, 15, 17, 18, 20, 22, 25, 26) were also found, but at a relatively low abundance (<5%).

The acidobacterial diversity patterns along a fire severity gradient were analyzed by the number of reads, number of OTUs, Shannon index and Simpson index. We found no significant differences in the same soil layer among different fire severities by one-way ANOVA (Fig. 3; Table S3). However, independent-sample $t$-tests showed a significantly lower number of reads, but higher Shannon and Simpson indices in the topsoil of the high-severity fire area compared with the subsoil ($P < 0.05$) (Fig. 3; Table S4).

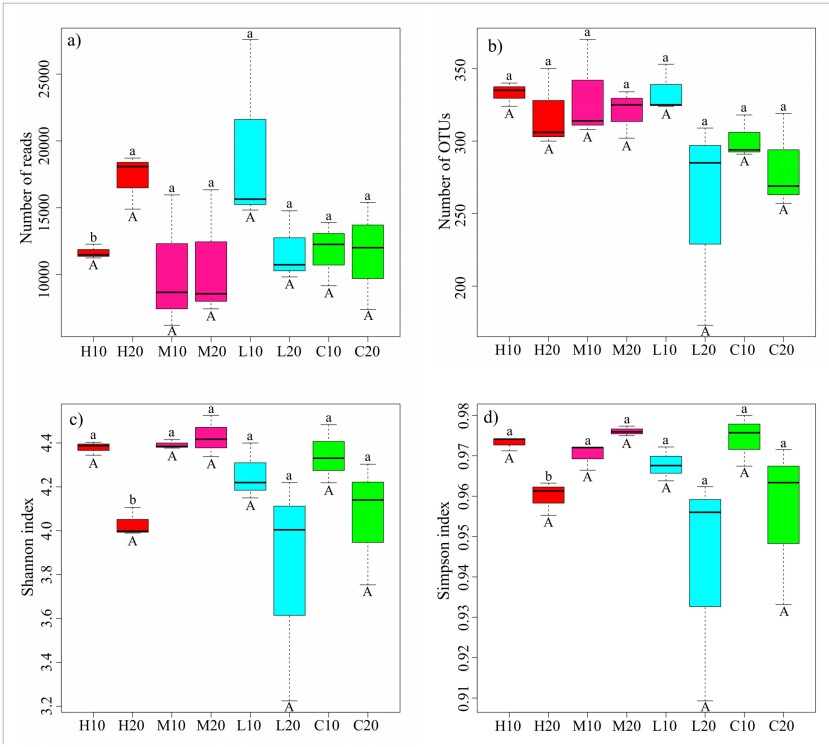

**Figure 3 Sequencing results and acidobacterial $\alpha$-diversity in different fire severity areas.** Different lowercase letters indicate a significant difference in different soil layers of the same fire severity plots at the $P < 0.05$ level; different capital letters represent a significant difference in the same soil layer of different fire severity plots at the $P < 0.05$ level. H, high severity; M, moderate severity; L, low severity; C, unburnt control; 10, topsoil (0–10 cm); 20, subsoil (10–20 cm).

NMDS analysis was used to distinguish differences in the acidobacterial community structure between the two soil layers. Figure S3 shows that the acidobacterial community structure of topsoil and subsoil is clearly divided into two parts. The PERMANOVA further confirmed that the acidobacterial community structure in the topsoil was significantly different from that in the subsoil ($R^2 = 0.1317$, $P = 0.048$).

## Relationship between the acidobacterial community structure and environmental factors

RDA was used to examine the relationships between the soil acidobacterial community structure and environmental factors. Based on the test of the variance inflation factor, soil TN, $NH_4^+$-N, pH, the diversity of understory vegetation ($H'_{Vegetation}$, $D'_{Vegetation}$ and $J'_{Vegetation}$), plant DBH, plant height and canopy density were selected for the RDA model of topsoil; and soil OM, $NO_3^-$-N, SM, pH, the diversity of understory vegetation ($H'_{Vegetation}$, $D'_{Vegetation}$ and $J'_{Vegetation}$), plant DBH and plant height were adopted for the RDA model of subsoil. Permutation test results indicated that the effects of soil pH, TN, $NH_4^+$-N, $H'_{Vegetation}$ and canopy density on the soil acidobacterial community structure reached a significant level in topsoil ($P < 0.05$) (Fig. 4A). Meanwhile, soil pH and OM were found to

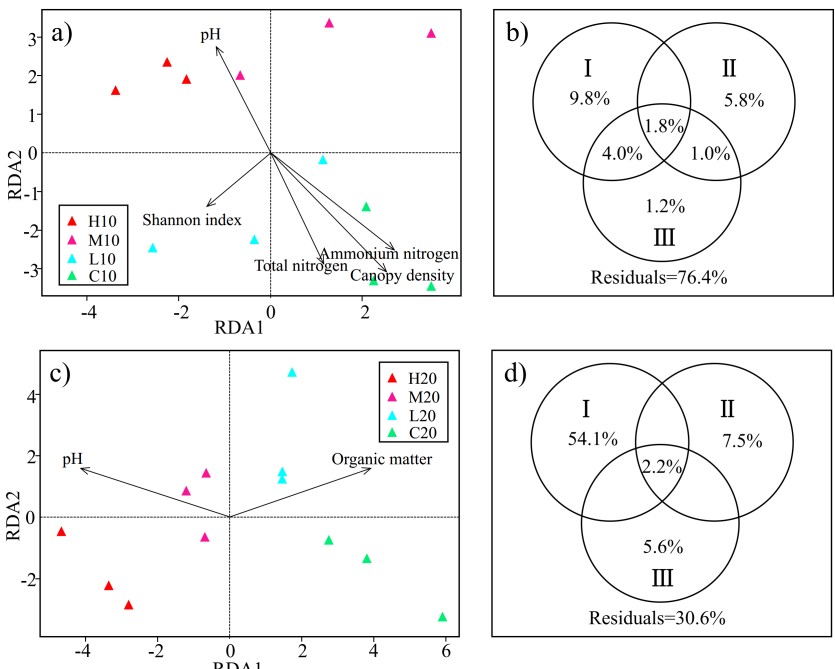

**Figure 4 Redundancy analysis (RDA) and variance partitioning analysis (VPA) show the relationship between acidobacterial community and environmental factors in the topsoil (A & B) and subsoil (C & D).** Circle diagrams indicate the independent and shared variance explained by environmental factors in the topsoil (B) and subsoil (D) respectively. I: edaphic factors; II: understory vegetation diversity factors; III: plant community structure factors. H, high severity; M, moderate severity; L, low severity; C, unburnt control; 10, topsoil (0–10 cm); 20, subsoil (10–20 cm).

be significant factors that influenced soil acidobacterial community structure in the subsoil (Fig. 4C).

On the basis of RDA, we classified the environmental factors selected by the model into three parts. TN, $NH_4^+$-N, pH, OM, $NO_3^-$-N and SM were classified as edaphic factors. H'$_{Vegetation}$, D'$_{Vegetation}$ and J'$_{Vegetation}$ were classified as understory vegetation diversity factors. Canopy density, DBH, and plant height were classified as plant community structure factors. VPA was conducted to assess the relative contributions of edaphic factors, understory vegetation diversity, and plant community structure to the acidobacterial community structure. In the topsoil, the combination of these variables explained 23.6% of the observed variation in soil acidobacterial community structure. Edaphic factors explained the largest fraction of the variation (15.6%), with an independent effect of 9.8%. Understory vegetation diversity and plant community structure independently explained 5.8% and 1.2% of the variation, respectively (Fig. 4B). In the subsoil, 69.4% of the variation was explained by the combination of the variables, with edaphic factors accounting for the largest fraction of the variation (56.3%) (Fig. 4D). The results showed that edaphic factors were the main factors affecting the acidobacterial community structure both in topsoil and subsoil. However, a large number of the factors involved especially in the topsoil, are still unknown.

Consistent with the results of the VPA, we also found different relationships between the relative abundance of *Acidobacteria* subgroups and soil factors. For example, in the topsoil, the relative abundance of *Acidobacteria* subgroup 1 was significantly positively correlated with $NH_4^+$-N ($r = 0.698$, $p < 0.05$), while the relative abundance of subgroup 7 was significantly negatively related to $NH_4^+$-N ($r = -0.655$, $p < 0.05$). In the subsoil, the relative abundance of *Acidobacteria* subgroups 1 ($r = 0.849$, $p < 0.01$) and 3 ($r = 0.607$, $p < 0.05$) were found to be significantly positively correlated with OM. By contrast, the relative abundance of subgroups 4 ($r = -0.630$, $p < 0.05$), 10 ($r = -0.577$, $p < 0.05$), 18 ($r = -0.651$, $p < 0.05$) and 20 ($r = -0.582$, $p < 0.05$) showed an opposite relationship with OM. We also found that not all *Acidobacteria* subgroups showed a consistent relationship with soil pH in the two soil layers. Except for *Acidobacteria* subgroups 3 and 4, the relationships between other subgroups and soil pH were found to be consistent in the two soil layers. However, the dominant subgroups 3 and 4 showed a negative relationship in the topsoil (subgroup3: $r = -0.059$; subgroup4: $r = -0.092$), but a positive relationship in the subsoil (subgroup3: $r = 0.025$; subgroup4: $r = 0.803$) (Tables S5; S6).

## DISCUSSION

Soil *Acidobacteria* are widespread and abundant in natural ecosystems (*Mbuthia et al., 2015*; *Liu et al., 2016*), and have been analyzed in different environments, such as in wastewater treatment processes (*Crocetti et al., 2002*), hot springs (*Hobel et al., 2005*), sand dunes (*Smith, Abed & Garcia-Pichel, 2004*), an acidic sphagnum peat bog (*Dedysh et al., 2006*), Amazon forest soils (*Navarrete et al., 2015*) and Chinese fir plantations (*Liu et al., 2017*). However, the responses of the acidobacterial community to a fire disturbance are still limited. In this study, we analyzed the variation of *Acidobacteria* in a *P. tabulaeformis* forest after a wildfire. The results of Illumina MiSeq sequencing showed that 31.77% of the reads belonged to *Acidobacteria* across all the soil samples. Overall, 18 different *Acidobacteria* subgroups were detected in this study, with subgroups 4, 6, 1, 3 and 2 found to be dominant. This finding is similar to that of *Zhang et al. (2014)*, who also found 19 known subdivisions in Shennongjia Mountain, Central China, of which subdivisions 6, 1, 2, 3, 4 showed relative high abundance.

Fire has a strong effect on the diversity of the soil microbial community. A decrease of soil bacterial diversity can be observed after burning, especially after a high-severity fire (*Ferrenberg et al., 2013*; *Xiang et al., 2014*). *Mabuhay, Nakagoshi & Isagi (2006)* reported that the burnt areas still showed very low microbial community diversity even 1 year after a fire. *Xiang et al. (2014)* observed that the diversity of the bacterial community returned to match that of the unburnt controls 11 years after the fire. Microbial diversity tends to recover gradually over a long time (*Shen, Chen & Lewis, 2016*). However, we found no significant differences of the acidobacterial α-diversity in the topsoil or subsoil (Fig. 3). We speculated that the soil acidobacterial community might recover in a much faster time. Interestingly, although the number of reads in the topsoil was significantly lower than that in the subsoil, a significantly higher acidobacterial α-diversity (i.e., Shannon and Simpson indices) was observed in the topsoil than in the subsoil of high severity plots

(Fig. 3). This phenomenon may be due to the edaphic and vegetable heterogeneity caused by fire, enabling bacterial diversity to be maintained and even improved with the change of microbial competition for resources (Chesson, 2000; Otsuka et al., 2008; Artz et al., 2009).

Numerous studies have demonstrated that soil microbial communities are closely related to aboveground vegetation and soil properties (Bardgett, 2005; Chu et al., 2016). Among them, Acidobacteria were found to play a role in promoting soil carbon cycling, as it can degrade complex plant derived polysaccharides such as cellulose and lignin (Huang et al., 2015). Soil pH has often been reported as a dominant factor in shifting acidobacterial communities (Xiong et al., 2012; Xiang et al., 2014; Chu et al., 2016). This conclusion has also been confirmed in this study, where we found that soil pH was a key factor both in the topsoil and subsoil (Fig. 4A; Fig. 4C). Previous studies have shown that different subgroups presented inconsistent correlations with respect to soil pH. For example, Turlapati et al. (2013) reported that subgroups 1, 2, 3, 12, and 13 were positively correlated with soil pH, but subgroups 4, 6, 7, 10, 11, 16, 17, 18, 22, and 25 showed a negative correlation. However, an opposite relationship was reported in the review article by Kielak and colleagues (Kielak et al., 2016), who showed that pH had a negative relationship with the abundance of subgroups 1, 2, 3, 12, 13, and 15, but a positive correlation with subgroups 4, 6, 7, 10, 11, 16, 17, 18, 22, and 25. In this study, we found not only that the relationships between Acidobacteria subgroups and soil pH were inconsistent in the same soil layer, but also that certain subgroups had different relationships with soil pH in different soil layers. For instance, the dominant subgroups 3 and 4 showed a negative relationship with soil pH in the topsoil, but a positive relationship in the subsoil (Tables S5; S6). We hypothesized that the varied responses of the Acidobacteria subgroups to soil pH represented the key determinant of the different relationships between Acidobacteria and soil pH. However, the relationship between Acidobacteria subgroups and soil properties with the change of soil depth still needs further study.

Soil nitrogen and OM were also found to be important parameters in driving the acidobacterial community structures in the topsoil and subsoil, respectively (Figs. 4A; 4C). Ward et al. (2009) reported that Acidobacteria played an important role in nitrogen cycling, and subgroup 1 had related correlations with nitrate and nitrite. In this study, we also found a significant relationship between subgroup 1 and $NH_4^+$-N in the topsoil (Table S5). Acidobacteria are usually considered to be oligotrophic (Fierer, Bradford & Jackson, 2007; Jones et al., 2009). Navarrete et al. (2015) found a significant positive relationship between the abundance of Acidobacteria and soil OM. Liu et al. (2016) also observed a larger abundance of Acidobacteria in high organic carbon content soils than in low carbon content soils in the black zone of northeast China. It has been pointed out that not all subgroups or OTUs of this phylum would be oligotrophic (Fierer, Bradford & Jackson, 2007; Chu et al., 2016). In this study, subgroups 4, 10, 18, and 20 were found to be significantly negatively related to soil OM in the subsoil which suggesting oligotrophy (Table S6). However, a significant positive relationship was also detected between the dominant subgroups 1 and 3 and soil OM in this soil layer (Table S6). Kielak et al. (2016) proposed that not all Acidobacteria possess the same ecological strategy, with a high variation of physiological characteristics within this phylum.

Extensive studies have suggested that there are strong interrelationships between aboveground vegetation and soil microbial communities (*Hart et al., 2005*; *Nielsen et al., 2010*; *Dangi et al., 2010*). A larger microbial biomass was found in a forest of greater aboveground diversity than in a homogeneous plant cover environment (*Fioretto et al., 2009*). *Hart et al. (2005)* considered that changes in vegetation community structure in the years following a fire might have a greater effect on soil microbial communities than the direct influence of the fire disturbance itself. For example, the soil nutrient and carbon content could be influenced by the inputs of litter (leaves, stems, roots) and exudate from newly established plants after burning (*Knelman et al., 2015*). In this study, the Shannon diversity and canopy density of aboveground vegetation were found to have significant effects on the acidobacterial community structure in the topsoil (Fig. 4A). We hypothesize that an important reason for the higher diversity of the acidobacterial community in the topsoil of the high-severity area is the increase in aboveground vegetation diversity (*Han et al., 2007*; *Zhang et al., 2014*) and we also found H'$_{Vegetation}$ play an important in driving the change of acidobacterial community in this soil layer (Fig. 4A). Further, changes in plant community structure, such as the decline in canopy density, increase the light availability and soil temperature in the forest, which are beneficial for the bacterial communities (*Arunachalam & Arunachalam, 2000*; *Bolat, 2014*). However, the result of the VPA showed that a large amount of the variation remained unexplained in this study, especially in the topsoil (Figs. 4B; 4D). Other studies show that climate change (*Maestre et al., 2015*), soil metal elements (*Navarrete et al., 2015*), and soil enzyme activities (*Zhang et al., 2014*) also play important roles in determining microbial community structures. Accordingly, further comprehensive studies of *Acidobacteria* are necessary.

## CONCLUSION

In summary, this study conducted the response of soil *Acidobacteria* to fire disturbance by the Illumina MiSeq sequencing technique. We found no significant differences in acidobacterial $\alpha$-diversity in the topsoil or subsoil across different fire severities. However, the community structure of soil *Acidobacteria* in the topsoil was significantly different from that in the subsoil. We found that soil pH was a key factor in driving acidobacterial community structure in both soil layers while several other factors (e.g., total nitrogen, NH4+-N, H'Vegetation and organic matter) also have an important effect. The *Acidobacteria* subgroups were found to show different variations after a wildfire. The differential responses of the *Acidobacteria* subgroups to specific environmental factors can help to reveal what drives their population changes in forest soils recently suffering a fire disturbance and open the possibilities to explore acidobacterial subgroups as bioindicators for ecosystem restoration in *Pinus tabulaeformis* stands.

### Funding
This work was supported by the National Nature Science Foundation of China (31770696), the Key Project of National Key Research and Development Plan (2017YFD0600106) and the College of Forestry/Beijing Key Laboratory of Forest Resources and Ecosystem Process, Beijing Forestry University. The funders had no role in study design, data collection and analysis, decision to publish, or preparation of the manuscript.

### Grant Disclosures
The following grant information was disclosed by the authors:
National Nature Science Foundation of China: 31770696.
Key Project of National Key Research and Development Plan: 2017YFD0600106.
College of Forestry/Beijing Key Laboratory of Forest Resources and Ecosystem Process, Beijing Forestry University.

### Competing Interests
The authors declare there are no competing interests.

### Author Contributions
- Weike Li conceived and designed the experiments, performed the experiments, analyzed the data, contributed reagents/materials/analysis tools, prepared figures and/or tables, authored or reviewed drafts of the paper, approved the final draft.
- Xiaodong Liu conceived and designed the experiments, authored or reviewed drafts of the paper, approved the final draft.
- Shukui Niu conceived and designed the experiments, performed the experiments, authored or reviewed drafts of the paper, approved the final draft.

### Data Availability
 The raw reads are available in the National Center for Biotechnology Information Sequence Read Archive database (accession number: SRP158101).

### Supplemental Information
Supplemental information for this article can be found online at http://dx.doi.org/10.7717/peerj.8047#supplemental-information.

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
