# Peer review of "Differential responses of the acidobacterial community in the topsoil and subsoil to fire disturbance in Pinus tabulaeformis stands"

_PeerJ, doi:10.7717/peerj.8047_

## Round 0.1 · original submission · Major Revisions

Two experts have reviewed the manuscript and agree that the response of soil microorganisms to fire disturbance is an important, yet understudied, line of research in the soil sciences - and that your manuscript provides meaningful findings to this field of inquiry. They have provided a detailed critique of your manuscript and offer a number of insightful suggestions/corrections. Based on these reviews, I have decided that the manuscript requires major revisions for publication in PeerJ. Please respond to each of the reviewer’s comments, point by point, and, in particular, consider the points below.

Both reviewers are concerned that insufficient background is provided to frame the research. Why focus on the Acidobacteria instead of the entire soil community? What are the known roles of Acidobacteria in soils – particularly forest soils? Greater detail on how the fire was designated as either low, moderate or high intensity is warranted. This information is critical to justifying the aims of your study. Also, the study uses a single collection point (6 months) post-fire from which the data and conclusions are drawn, thus it is speculative to discuss results in terms of recovery.

Both reviewers also provide sound recommendations concerning the statistical approaches used for data analysis throughout the manuscript. Note also that the number of reads of a given OTU or taxa per sample is not informative unless the dataset is normalized (to an equal number of reads per sample) presented as a relative abundance.

A paragraph or section summarizing the significance (take home message) of the study should also be added to the end of the manuscript.

Reviewer 1 ·

Basic reporting

Please, include “Conclusion” to summarized the results and suggests the meaning of study or further study. Some citations are inappropriate, so please check again as mentioned below.

Experimental design

Experimental design was well designed, while the rationale of the study to focus on Acidobacteria is insufficient. I understand that Acidobacteria is abundant in soil and has important roles in soil ecosystem. However, other bacteria (e.g. Proteobacteria) are also abundant and have some roles in forest. Why other bacteria are not need to be studied? It is also important in soil ecosystem. Comparing responses of different bacterial phyla depend on wildfire severities can be interesting subject. Covering all bacteria is not essentially required, but authors should provide the reason to focus on Acidobacteria in this study.

Validity of the findings

Results are interesting, but some statistical methods are concered. All statistical tests in this study are parametric analysis, But, boxplots in Fig. 2 looks non-normal distribution and there is none of mention about testing assumptions for parametric test (e.g. normality test). Is it tested appropriately? Then please mention about it in the manuscript.

Additional comments

# Major comments
1. Rationales of study is needed as mentioned above.
2. Authors used Bray-Curtis dissimilarities on ordination analysis. It is not considered phylogenetic relationships of the species (OTUs) in communities, thus it can miss some aspect of natural community structures. Unifrac dissimilarity is recommended to be included additionally in analysis.
3. The abundance of other bacterial phyla is not provided. Is Acidobacteria most abundant in all samples? Although Acidobacteria is main focus in this study, supplementary information for other phyla is informative for readers.

# Minor comments
Acidobacteria is italicized in all part of the manuscript. But, the names of higher taxonomic levels (family, order, class, phylum or division, and kingdom) should not be italicized.

Line 94: Please provide the reference of classification standard.
L126-127: Citation of primers is not relevant. Please provide references appropriately.
L144-145: This sentence is repetition of following sentence (L14-146).
L146: The meaning of “300 bp reads” is ambiguous.
L152: Citation is wrong (UCLUST).
L153: please provide citation for Usearch (Edgar 2010).
L153-154: What kind of algorithm was used for chimera detection. De novo or reference based?
L154-156: Is it correct to use UCLUST for taxonomic assignment? Then, what is the meaning of a confidence threshold?
L155: Wrong citation for SILVA.
L164-166: What is the meaning of relationship? Diversity or abundance?
L179-180: Citation of vegan package and R. Please use a command “citation()” or “citation(“vegan”)” on R.
L190-195, 201-203: Interpretation of the results is not appropriate in Results parts. Please move to Discussion part.
L254-255: What is the meaning of close relationship? Between what?
L272: P. tabulaeformis forest is better.

Figure 2. Insufficient legend (“different capital letters ?”)
Figure 3. Please change the color of M10 for better recognition. The legend of 3b is incorrect (e.g. H10 > H20).
Figure 4. A-F in venn diagram are not need in this Figure. It is no meaning and distractive.

Reviewer 2 ·

Basic reporting

The manuscript is generally written in clear English and conforms to professional standards. The article is well-organized, utilizing a consistent structure throughout the paper. The Materials and Methods, and Results sections in particular are very well organized and written with clear, direct language that facilitates understanding of the primary research findings. I also appreciate the inclusion of Table 1, which provides information about the edaphic variables in the research plots, and Table S2, which provides information about the vegetation structure. This information provides valuable ecological context, and I commend the authors for including it.

The primary weaknesses of the paper are that it contains insufficient background information to properly frame the research topic, and several of the figures require revision. Specifically, the Introduction and Discussion sections should be revised to more thoroughly review the current state of knowledge on the impacts of fire on the soil bacterial community. The authors should provide more detail about the role of Acidobacteria in ecosystems, and explain why the community structure of Acidobacteria sub-groups is important to ecosystems. For example, in lines 41-55 the authors briefly discuss the prevalence of Acidobacteria in soil environments, but the role of Acidobacteria in general, and specific Acidobacteria sub-groups on ecosystem processes is not discussed. It is thus unclear why the impact of fire on Acidobacteria is important to the ecosystem, and why the authors focus on the community structure of the various subgroups of Acidobacteria.

Additionally, the authors assess the strength of various edaphic and environmental variables as drivers of Acidobacteria community structure, including organic matter concentration, nitrate-nitrogen, soil moisture, pH, vegetation diversity, and tree structure. However, only background information regarding the role of pH in driving Acidobacteria community structure is provided in the introduction. Justification is not provided for the inclusion of the other variables in this study, and the manuscript should be revised to review the state of knowledge of these variables as drivers of Acidobacteria activity and community structure. Similarly, in the discussion, the authors describe environmental factors that appear to be driving changes in Acidobacteria communities, but the downstream effects of these changes is not given sufficient consideration. In both the Introduction and Discussion sections, the authors frequently make vague, generalized statements without providing precise, detailed examples in support (see specific comments below). The authors should revise to provide literature-based support of their claims.

Nearly all figures and tables require revision to formatting (mostly text size): see Specific Comments in a following section. Please provide an additional figure showing photos of the unburned and burned areas: this is very important for helping your reader see the visual impacts of the fire, associated with each severity level you classified.

Lastly, it is unclear why the authors focus solely on Acidobacteria after sequencing the entire bacterial community. Why did the authors focus only on Acidobacteria and not include other ecologically important and abundant phyla like Proteobacteria or Bacteroidetes? The approach should be explained and justified in the introduction. Please also include a conclusions section or paragraph to clarify the take-away message from your study, and the implications for this research field.

Experimental design

The manuscript describes primary research within the aims and scope of PeerJ journals. The manuscript addresses three research questions that are clearly identified and prominent in the introduction, and these questions are clearly addressed by the results. The analytical methods utilized are described clearly and with sufficient detail, are technically sound and reproducible, and are within current methodological standards.

The paper would primarily benefit from providing some additional information about the experimental design, and site and soil characteristics. Specifically, the spatial distribution of the research plots needs clarification. For example, it is unclear whether the three plots within each severity level were located in a single area of contiguous burn severity, or whether the plots are located in separate severity “patches.” This should be clarified in the main text, and/or figure S1 should be revised to illustrate the pattern of burn severities across the research area and the distribution of the research plots. The mean and minimum distances between research plots should also be reported. Additionally, the authors should provide more information about the distribution of soil types across severity levels, and information about pre-fire vegetation structure. This would provide important context for the patterns of Acidobacteria community structure and organic matter content observed, and aid in interpreting whether these results are related to pre or post-fire conditions. If possible, representative photographs from each severity level should be included, which would provide context about pre and post-fire vegetation structure, and allow visualization of the fire’s impact on the soil.

The investigators clearly identify the knowledge gap they are investigating: to understand the impact of fire severity on the Acidobacteria community in soil. However, the relevance and importance of this topic is not given sufficient attention in the introduction, discussion, or conclusion sections of the manuscript, and additional information should be provided to justify the study (see comments in the basic reporting section above). The authors should also more directly state how their results contribute to filling this knowledge gap by including a conclusions section.

Validity of the findings

The investigators used straight-forward statistical techniques. Some minor clarification in the text is needed about how they implemented their t-tests when comparing different soil layers (see specific comments below). Although the authors used one-way anovas to evaluate differences among severity levels within soil depth increment, followed by t-tests to determine differences between depth increments within severity level, this could also be treated as a two-way anova followed by protected pairwise comparisons. The authors included appropriate controls, and the level of replication is acceptable, but additional information is needed about the location of the research plots relative to each other to ensure pseudo-replication is not a concern (see also comments in “Experimental Design” section above). The authors should revise the manuscript to include f- & p-values and t-statistics in the text, figures, and/or tables. Lastly, the authors should include a conclusions section that connects to the original question investigated.

Some important questions to consider relate to:
1) the scale of the severity assessments, and 2) the fact that the severity estimates were based on whole-stand characteristics and provide relatively low resolution of effects on the soil. For example, severity was assessed at a 20x20 m plot scale, but we know that there is a lot of spatial variability in fire effects. The authors should discuss whether 5 soil subsamples really represents a 20x20m area. Was severity totally consistent within each plot? Severity level may be more representative of effects on soil if it is assessed at the soil sample location(s). The data from this study also shows that fire may only affect soil bacteria in high-severity burns where all the organic matter has been consumed.

Additional references that may help strengthen the discussion include:
Hartford and Frandsen. 1992. When It's Hot, It's Hot... Or Maybe It's Not! (Surface Flaming May Not Portend Extensive Soil Heating). International Journal of Wildland Fire 2:139-144.
Kolka et al. 2014. Post-fire comparisons of forest floor and soil carbon, nitrogen and mercury pools with fire severity indices. Soil Sci. Soc. Am. J. 78:S58-S65
Kolka et al. 2017. Emissions of forest floor and mineral soil carbon, nitrogen and mercury pools and relationships with fire severity for the Pagami Creek Fire in the Boreal Forest of northern Minnesota. International Journal of Wildland Fire 26:296-305

Additional comments

Dear Authors,
This is a very interesting and novel study that provides important information about the effects of fire and burn severity in an understudied forest type. The specific comments below identify ways to help clarify the importance and value of your study.

Specific Comments:
Abstract:
Line 2: Many researchers consider the response of soil bacteria to fire to be a significant knowledge gap in the field. For example, see Pressler et al. 2018 (doi: 10.1111/oik.05738).
Line 10: How does the abundance of the Acidobacteria compare to other phyla? Was this the most abundant phylum?
Line 22: 15.6% is not “most of the variation.” Please revise for clarity.
Lines 23-23: What are the large gaps that still need to be explored? Specifically identifying these gaps will be a more effective concluding sentence for the abstract.

Introduction:
Line 33: What do you mean by “the balance of the ecosystem” and how do microorganisms regulate this balance? Can you also provide examples regarding the “crucial and unique role in energy flow and circulation of material?”
Line 37: The role of fire severity on microbial community structure and activity remains an open question. For example, Pressler et al. (2018) concluded that there are insufficient studies to draw strong conclusions about the role of fire severity on the bacterial community. Furthermore, one of the studies cited here (Weber et al. 2014) found inconsistent response of microbial communities to fire severity and the other (Hamman et al. 2007), found differences in microbial community structure related to fungi, not bacteria. This portion of the introduction could be rewritten to highlight the uncertainty that exists regarding bacteria response to fire severity and improve the justification for your study. Either way, this sentence should be revised to clarify that the impacts of fire severity on bacterial communities is far from a settled question.
Line 40: Please discuss why the effects of wildfire on specific bacteria phyla are important.
Line 44: Do the different Acidobacteria subgroups have different ecological roles? Here, it is important to explain why the structure of Acidobacteria community matters for the ecosystem, and thus, why your research is important.
Lines 49 – 51: Please explain how the oligotrophic characteristics of Acidobacteria affect ecosystem processes, define what eosinophilia is, and please be more specific about the phylogenetic diversity of the phylum.
Line 51: Delete “it should be pointed out that” from this sentence.
Line 51-55: How might the tolerance of Acidobacteria to different environmental conditions affect their response wildfire or ecosystem function after a fire? Do different sub-groups consistently show different responses to various environmental conditions?
Line 57: Please provide a reference of the statement that wildfire has especially variable results on Acidobacteria, or revise this sentence. I am skeptical that there is enough research available to draw conclusions about the impacts of wildfire on Acidobacteria relative to other phyla.
Lines 64-66: How does Acidobacteria vary in response to “sampling time, location, fire severity, and vegetation type” that leads you to this conclusion? How do the biological characteristics of Acidobacteria subgroups vary and how do these impact the response? Please provide additional detail and more references here.
Line 68-69: Why is knowledge of the response pattern at the subgroup level important? Again, this information is critical to justify this study.
Lines 70-72: These two sentences are out of place here and should be moved to the site description section of the Materials and Methods.

Methods:
Lines 86-87: Please provide more information about these soils. “Brown soil” and “cinnamon soil” will not be familiar to an international audience and more information about the characteristics should be provided here. At a minimum, please provide information about the typical textures, parent materials, horizonation, and pH of these soils.
Lines 91-92: The sentence “Soil samples were collected…” is out of place here. It could be moved to line 96, prior to “In each plot, five random soil cores…”
Line 94: Please provide a reference for the severity classification standard.
Line 94: Please include Table S1 in the primary text, not as a supplementary figure.
Line 94: Was severity classified for the entire 56.33 ha area that was burned? If so, what was the proportion that burned at high, moderate, and low severity? If not, how much of the burned area was assessed for severity?
Lines 94-95: Were the three plots in each burn severity area located in a single contiguous area of burn severity, or were they located in distinct burn severity patches that were separated by regions of variable burn severity?
Line 96: “Shady” is not an aspect, please provide the compass direction of the aspect.
Line 160-162: Is it correct that two separate ANOVAs were conducted, one for each soil layer? Please clarify.
Line 163-167: Was a single t-test performed (between depth increments, across all severity levels), or were separate t-tests performed to compare depths, within each severity level?

Results:
Line 185-186: The large difference in OM content at the high severity level, especially in the 10-20 cm layer is interesting, and requires further investigation and explanation. Although losses in soil OM following fire are not unusual, these losses are usually restricted to the upper few centimeters, because the heat pulse from fire generally does not penetrate very far into the mineral soil. This makes me suspect that the differences in OM content existed prior to the fire, and by extension, the observed differences in Acidobacteria community in the high severity plots may have existed prior to the fire. Is there any pre-fire soil data available from these plots? Is the soil type the same in the high severity plots as the other plots? Was the pre-fire plant community composition and structure similar to the other plots? At the very least, you should provide representative photographs from each severity level so the reader can assess whether the pre-fire plant structure was similar, and you should provide more background information about the soils. Please see the following for information about the impacts of fire on soil and soil OM:
Nave et al. 2011 (Ecological Applications 21(4), p 1189-1201),
Certini et al. 2005 (doi: 10.1007/s00442-004-1788-8),
Gonzalez-Perez et al. 2004 (doi: 10.1016/j.envint.2004.02.003)
Adkins et al. 2019 (doi: 10.1016/j.geoderma.2018.07.009).

Line 186-195: Please provide the specific f-values and p-values for each variable assessed with ANOVA. At a minimum, please also include p-values for each variable in Table 1.
Lines 190-195: This speculation about the reasons for differences in SM belongs in the discussion section, not the results.
Lines 192-194: Increases in SM may also be due to reduced uptake of soil water by plants.
Line 199: Please provide specific f- and p-values for each variable here and in Figure S2 (or put the f-statistics into a table).
Line 199-200: This sentence is redundant with the preceding sentence. Please delete.
Line 207: How does the proportion of reads belonging to Acidobacteria compare to other abundant groups? Was Acidobacteria the most abundant phylum?
Line 220: Please report specific f-values and p-values here and in Figure 2.
Lines 254-263: Please report r2 values and p-values for each correlation referenced in this paragraph.

Discussion:
Line 269: Please provide specific examples of the different environments.
Line 271: In the introduction you cite papers regarding the response of Acidobacteria to fire, but here you say the response in unknown. Please clarify what the knowledge gap is in regards to Acidobacteria response to fire here and in the introduction.
Line 273: Again, please clarify how the abundance of Acidobacteria compares to other phyla. Why is no information about other dominant phyla provided in this manuscript?
Line 276: Please summarize Zhang’s et al. (2014) findings and clarify how it compares to your results.
Line 278-280: You claim “a large number of studies have shown microbial diversity decreases significantly,” but only cite two studies here. Again, please see Pressler et al. 2018, who suggest there are not enough studies on bacterial communities post-fire to draw strong conclusions.
Line 285: Your findings are not consistent with these results. A recovery within 6 months is much faster than the recovery in 11 years observed by Xiang et al. (2014). Furthermore, your study assessing changes after a single wildfire is not comparable to Shen’s et al. study which assessed the response to repeated prescribed fires. Please update this section with more relevant references and clarify the differences in your results compared to other.
Lines 286-287: Because you do not have pre-fire and immediate post-fire Acidobacteria data, you cannot make the claim the Acidobacterial community recovered in 6 months. More likely is that the Acidobacteria community was not affected by the low and moderate severity fires. Most studies have indicated that it takes several years for the microbial community to recover, suggesting its unlikely that the Acidobacteria community recovered in 6 months.
Lines 289-290: Please revise this sentence to clarify that alpha diversity was higher in the topsoil than the subsoil in high severity plots, or whether it means that the alpha diversity was higher in topsoil in high severity plots than in the topsoil of the other plots.
Line 290 -293: Please provide additional detail about how edaphic and vegetable heterogeneity affects microbial competition for resources, and how this in turn impacts bacterial diversity. How might these processes impact Acidobacteria specifically?
Lines 294-295: Please summarize/provide cited examples about the ways in which microbial communities are impacted by aboveground vegetation and soil properties, and how fire impacts these properties.
Lines 296-297: Please explain how and why Acidobacteria promote nutrient cycling, plant growth, and soil restoration after disturbances. How does this differ from other bacteria phyla? Do different Acidobacteria subgroups play different roles?
Line 318: What specifically is the role that acidobacteria play in the nitrogen cycle?
Lines 321-330: In this section, please indicate that you are suggesting that a negative correlation with OM suggests oligotrophy.
Lines 321-330: Organic matter quality (e.g. carbon to nitrogen ratio) plays an equally or more important role as OM content in the colonization of soils of oligotrophs versus copiotrophs. For example, if OM is of low quality (e.g. high carbon to nitrogen ratio), we would expect to see relatively more oligotrophs regardless of OM content. You have not included data on OM C:N here, but you should at least mention the potential role of OM quality.
Lines 339-340: This sentence is redundant with the preceding portion of the paragraph. Please delete.
Lines 342-344: Consider assessing this hypothesis with your data
Lines 346-347: Please specify how these factors are beneficial for bacterial communities and how they might impact the differences you observed.
Lines 349-352: This is a weak conclusion to your paper, which presents an interesting & novel study. Please revise the text to conclude by highlighting your own findings, not the findings of others. Emphasize how your study adds valuable information to our understanding of fire in ecosystems around the world, and especially how it advances knowledge about fire's effects on forest soil. Make sure these are clear for your reader, so add this text in a separate conclusion section, or even in a conclusions paragraph.

Tables & Figures
Table 1: Please replace “Indices” with something more descriptive like “soil properties.”
Table 1: The soil depth column lists specific depths, but in the manuscript body you refer to these as “topsoil” and “subsoil.” Please use labeling in the tables that are consistent with the body text.
Table 1: Please spell out the full fire severity label rather than using abbreviations.
Table 1: Please add a column displaying the f-values and p-values for your ANOVA results.
Figure 1 Legend: Please also specify the forest type, time since fire, and define the abbreviations used in the x-axis.
Figure 3 Legend: Please define the abbreviations used in the figures and specify the different depths of topsoil and subsoil.
Figure 2: The figure text sizes should be greatly enlarged. Please label the panels as a, b, c, and d. Move the panel titles to become the y-axis labels.
Figure 3: The figure text sizes should be greatly enlarged. The legend for panel b includes the incorrect labels.
Figure 4: I recommend moving the Figure 4 panels into Figure 3 as panels c and d (or into insets in panels a and b), and combining the captions. Consider following the format used by Robert et al. 2018 in Ecography 40:1-16, doi: 10.1111/ecog.03553 (https://onlinelibrary.wiley.com/doi/abs/10.1111/ecog.03553)

Supplementary Tables and Figure Captions: all captions should be revised to provide much more detail. They should all be written similarly to what is provided in the main manuscript figures and tables: remember that a reader should be able to understand & interpret each figure and table as a “stand-alone” unit, without the reader needing to rely heavily on the manuscript text.
Table S1: Please spell out the fire severity level rather than using single-letter abbreviations. Cite your source in the caption to clarify where this severity classification system came from (it looks similar to one by Ryan & Noste, but if you modified a published severity classification system, then please provide a clear rationale for how & why it was modified. If you used a published severity classification system, then just report the original classification descriptions for each category…the current descriptions in this table are very unclear. Please revise the criteria description for the moderate severity category to be more descriptive. Use the term “low-severity,” not “low-grade.” Revise “height” in the moderate severity description to read “high severity.” Capitalize “unburned” in the control description. Please include this table in the main manuscript rather than as supplementary information. What does “burned down” mean? Do you mean consumed? Charred but still present?
Table S2: Please spell out the fire severity level rather than using abbreviations. Please change the column heading “Burning degree” to follow the terminology used in other tables & the text.
Figure S1: Add a scale bar. Also please add another panel to this figure to show the location and distribution of your research plots within your study site.

---

## Round 0.2 · Minor Revisions

Thank you for taking the time to respond to the reviewer's initial comments and submitting a much improved manuscript. The reviewers still have several concerns that could be quickly addressed.

First, both agree that several sections of text currently in the Results should be moved to the discussion section or deleted altogether. See comments from Reviewer #1 for details.

Second, Reviewer #2 is still concerned that the introduction does not contain enough background information concerning the known/potential roles of Acidobacteria (and its various subgroups) in soils. Reviewer #2 (and the editor) believe inclusion of this information should be straightforward and would greatly improve the justification and interpretation of the current study for readers not versed in the Acidobacteria literature. Several recent publications should provide a solid start: 1) Belova et al. “Hydrolytic Capabilities as a Key to Environmental Success: Chitinolytic and Cellulolytic Acidobacteria From Acidic Sub-arctic Soils and Boreal Peatlands”. Front Microbiol. 2018; 9: 2775; 2) Hausmann et al. “Peatland Acidobacteria with a dissimilatory sulfur metabolism”. The ISME Journal volume 12, pages1729–1742 (2018); 3) Eichorst et al. “Genomic insights into the Acidobacteria reveal strategies for their success in terrestrial environments” Environmental microbiology 20, no. 3 (2018): 1041-1063; 4) Diamond et al. “Mediterranean grassland soil C–N compound turnover is dependent on rainfall and depth, and is mediated by genomically divergent microorganisms”. Nat Microbiol. 2019 May 20. doi: 10.1038/s41564-019-0449-y. In particular, the fourth reference provides new exciting evidence that acidobacteria play important roles in C cycling in soils.

Finally, Reviewer #2 provides additional minor suggestions that should be addressed.

These changes should require very limited literature search/reading and a small amount of writing, but would result in a manuscript ready for publication.

Reviewer 1 ·

Basic reporting

Good.

Experimental design

Good.

Validity of the findings

Good.

Additional comments

The manuscript has revised well.

L185-189, 195-197: I don't agree with the response from the authors. If you don't want to make a special discussion or cannot find the place to put it, it means that this contents are not needed in this manuscript, thus it can be removed. If you have something to discuss, please do it in Discussion.

"So we don’t make a special discussion about the soil and vegetation characteristics in the discussion part. We've only made a brief discussion of the interesting phenomena in this part of the results. Besides, we haven’t found a suitable place in the discussion part to put this content."

Reviewer 2 ·

Basic reporting

The revisions the authors have made to their manuscript have improved the quality and clarity of the paper. The revisions to the methods, results, and discussion sections are generally well-done and satisfy most of my comments. The figures have been adequately revised for the most part, and the inclusion of photographs of the severity levels are very helpful. However, the paper still does not include sufficient background information to justify the study. This weakness is reflected by inadequate response of the authors to many of the changes suggested for the introduction. The authors still need to provide more detail about the role of Acidobacteria in ecosystems and explain why the community structure of Acidobacteria is important. It remains unclear why the impact of fire on Acidobacteria sub-groups is of ecological significance. Additionally, as mentioned in my previous review, the authors assess the strength of various edaphic and environmental variables as drivers of Acidobacteria community structure, including organic matter concentration, nitrate-nitrogen, soil moisture, pH, vegetation diversity, and tree structure. However, only background information regarding the role of pH in driving Acidobacteria community structure is provided in the introduction. Justification is not provided for the inclusion of the other variables in this study, and the manuscript should be revised to review the state of knowledge of these variables as drivers of Acidobacteria activity and community structure.

Experimental design

The authors have sufficiently addressed most of my comments regarding experimental design. However, more information is still needed regarding the spatial distribution of sample plots within the research area. For example, it is still unclear whether the three plots within each severity level were located in a single area of contiguous burn severity, or whether the plots are located in separate severity “patches.” The authors also did not provide information about the mean and minimum distances between research plots as I requested. Figure S1 still needs to be revised to include the locations of sample plots.

Validity of the findings

The authors have provided additional information about statistical methods and results, improving confidence in the validity of their findings. In a few instances, some additional information needs to be incorporated into the manuscript text (detailed in specific comments below) to aid in reader interpretation of results. Finally, as previously requested, and as detailed in the experimental design section above, more information is needed about the location of the research plots relative to each other to ensure pseudo-replication is not a concern. This information should be provided both in the materials and methods section of the manuscript and by revising Fig S1 to include more information.

Additional comments

Specific Comments:
The authors adequately addressed most of the issues I identified in the specific comments section of my previous review. Below, I provide follow-up responses to the many that have not been adequately addressed by the authors. Between lines beginning XXXX I have provided (1) my original reviewer comment, (2) the authors’ response, and (3) my follow-up response.

Original Reviewer Comment: Line 22: 15.6% is not “most of the variation.” Please revise for clarity.

Author Response: we rewrote this sentence. The revised sentence is “A variance partitioning analysis (VPA) showed that edaphic factors explained the most variation both in the topsoil (15.6%) and subsoil (56.3%). ” (Please see line 21 in the revised manuscript).

Reviewer follow-up: Please revise to “...edaphic factors explained the highest relative amount of variation in both the topsoil (15.6%) and subsoil (56.3%).

XXXX
XXXX
Original Reviewer Comment: Line 33: What do you mean by “the balance of the ecosystem” and how do microorganisms regulate this balance? Can you also provide examples regarding the “crucial and unique role in energy flow and circulation of material?”

Author Response: “maintenance the balance of the ecosystem” means soil microbes perform certain functions to maintain the ecosystem stability, such as organic matter decomposition and driving soil nutrient cycling. At the same time, these physiological activities are also accompanied by material circulation and energy flow.

Reviewer follow-up: Please revise the manuscript text to clarify this to the reader. I suggest revising the sentence to read something similar to the information you have provided in your response to my original comment. For example, “As an important part of the forest ecosystem, soil micro-organisms maintain ecosystem stability through the role in driving organic matter decomposition and soil nutrient cycling.”

XXXX
XXXX

Original Reviewer Comment: Line 37: The role of fire severity on microbial community structure and activity remains an open question. For example, Pressler et al. (2018) concluded that there are insufficient studies to draw strong conclusions about the role of fire severity on the bacterial community. Furthermore, one of the studies cited here (Weber et al. 2014) found inconsistent response of microbial communities to fire severity and the other (Hamman et al. 2007), found differences in microbial community structure related to fungi, not bacteria. This portion of the introduction could be rewritten to highlight the uncertainty that exists regarding bacteria response to fire severity and improve the justification for your study. Either way, this sentence should be revised to clarify that the impacts of fire severity on bacterial communities is far from a settled question.

Author Response: Thank you very much for your correction. Our original description is not rigorous and accurate. I wonder if I could quote the red part in your question. I think your summary is very concise and representative. If you allow me to quote your summary, please see lines 35-39 in the revised manuscript.

Reviewer follow-up: Yes, it is OK to use reviewer language to help clarify a manuscript. I recommend deleting the portion of the sentence that reads “… and Hamman et al. (2007) reported differences in microbial community structure related to fungi, not bacteria” since fungi is not a focus of your study. You could perhaps expand on this point to highlight the fact that most severity studies focus on fungi, rather than bacterial communities, but in its current form, this portion of the sentence is out of place.

XXXX
XXXX


Original Reviewer Comment: Line 44: Do the different Acidobacteria subgroups have different ecological roles? Here, it is important to explain why the structure of Acidobacteria community matters for the ecosystem, and thus, why your research is important.

Author Response: Yes, different Acidobacteria subgroups have different ecological roles. This view is supported by the cited literature (Kielak et al., 2016) in my manuscript. Actually, I don’t think it is necessary to develop a description about the structure of Acidobacteria community and ecosystem. Firstly, in Kielak’s review article they have made a relatively comprehensive introduction about the Acidobacteria physiological and ecological characteristics. Secondly, our research focuses on the changing pattern of Acidobacteria subgroups to wildfire disturbance and tries to find the main driving factors. Moreover, we have made an emphasis that knowledge of the response pattern (at the subgroup level) after a fire disturbance is still limited. However, the ecological functions of Acidobacteria subgroups in ecosystem are really important. So we make a discussion about the relationship between Acidobacteria subgroups and environmental factors.

Reviewer follow-up: This response is not satisfactory. Providing a description of the ecological roles of the Acidobacteria sub-groups is important for justifying this study. Simply providing a literature reference without summarizing the ecological roles and importance is not sufficient. In its current state, the introduction still does not provide adequate justification for this study. Please add a paragraph explain what is known about the ecological roles and characteristics of Acidobacteria sub-groups.

XXXX
XXXX


Original Reviewer Comment: Lines 49–51: Please explain how the oligotrophic characteristics of Acidobacteria affect ecosystem processes, define what eosinophilia is, and please be more specific about the phylogenetic diversity of the phylum.

Author Response: in lines 49-51, I didn’t say the oligotrophic characteristics of Acidobacteria will affect any ecosystem processes. However, the oligotrophic characteristics of Acidobacteria are indeed closely related to some ecosystem processes, such as the decomposition of organic matter and carbon cycling. But here I mainly want to bring out the following content that although Acidobacteria exhibits oligotrophy, not all subgroups are oligotrophs. So it is necessary to get a further insight into this phylum on the subgroup level.
Eosinophilia is an inaccurate description. We deleted it.
Liu et al. (2016) have made a more specific report about the phylogenetic diversity of the Acidobacteria. We just quote his viewpoint here.

Reviewer Follow-up: As you acknowledge in your response, oligotrophic characteristics are related to ecosystem processes. This is important context for your study, so please add the information provided in your response to the manuscript text. Furthermore, this portion of the manuscript does not mention that not all subgroups are oligotrophic. Please add this information to this portion of the manuscript, along with appropriate citations.

XXXX
XXXX


Original Reviewer Comment: Line 51-55: How might the tolerance of Acidobacteria to different environmental conditions affect their response wildfire or ecosystem function after a fire? Do different sub-groups consistently show different responses to various environmental conditions?

Author Response: In lines 57-65 of the original manuscript, we had made an introduction about the different responses of Acidobacteria to fire disturbance. We speculated that in addition to objective factors, the different biological characteristics of the Acidobacteria subgroups were important factors leading to these inconsistent changes. Actually, previous studies found that different subgroups showed different responses to various environmental conditions. For example, Zhang et al. (2014) reported that the elevation gradient had different effects on Acidobacteria subgroups. The relative abundance of subgroup 1, 2, 3 increased with the altitude, and the relative abundance of subgroup 4, 5, 6, 16, 17 decreased with the altitude (Zhang et al., 2014). Furthermore, several researches also found that the effects of soil pH on Acidobacteria subgroups are not exactly the same (Jones et al., 2009; Turlapati et al., 2013; Kielak et al., 2016).
All these references can be found in the manuscript.

Reviewer Follow-Up: The specific information you provide in your response about how different sub-groups vary with environmental conditions is important context for your study that should be incorporated into the manuscript text. Although you provide the references in the manuscript, you could incorporate much more of the important information found within these references into the manuscript text.

XXXX
XXXX


Original Reviewer Comment: Lines 64-66: How does Acidobacteria vary in response to “sampling time, location, fire severity, and vegetation type” that leads you to this conclusion? How do the biological characteristics of Acidobacteria subgroups vary and how do these impact the response? Please provide additional detail and more references here.

Author Response: this conclusion is based on the literature I have read. According to previous studies, we speculate that the different changes of Acidobacteria after burning are due to two reasons. One is objective reasons, such as sampling time, research location, different fire severity, and different vegetation types. The other is the biological characteristics of Acidobacteria subgroups themselves. As I mentioned in my previous answer different Acidobacteria subgroups showed different responses to various environmental conditions (Zhang et al., 2014; Jones et al., 2009; Turlapati et al., 2013; Kielak et al., 2016). However, these studies are all not related to fire. Shen et al. (2016) found minor change of Acidobacteria phylum, but a significantly lower abundance of subgroup 1 after burning. So there must be other subgroups increase after fire. Why are the changes in subgroups of the same phylum different? Combined with previous researches, I think it may be because different Acidobacteria subgroups have different biological characteristics. According to our result, we did find that the relative abundance of Acidobacteria subgroups vary inconsistently after fire (Fig.2). Correlation analysis also showed that the relationships between Acidobacteria subgroups and environment factors are inconsistent (Table S5; Table S6). I think these findings will help us to get a deeper understanding of Acidobacteria.

Reviewer Follow-up: You have not revised the manuscript to explain how Acidobacteria vary in response to “sampling time, location, fire severity, and vegetation type.” Some of information you provided here and in previous responses (e.g. variability in subgroup abundance with elevation) is relevant to addressing this issue, but this and additional information regarding how time, fire severity, and vegetation type influence Acidobacteria communities still needs to be added to the manuscript. If this information has not adequately been assessed by previous studies, you should discuss this in your introduction and explain the role your research plays in filling this knowledge gap.

XXXX
XXXX


Original Reviewer Comment: Line 94: Please provide a reference for the severity classification standard.

Author Response: Thank you for your reminder. I forgot such an important reference. I have added a reference in the revised manuscript.
[1] Zheng Q., Cui, X. Y., and Di, X. Y. (2012). Effects of Different Forest Fire Intensities on Microbial Community Functional Diversity in Forest Soil in Daxing’anling. Scientia Silvae Sinicae. 48, 95-100.
Reviewer Follow-up: Please also provide this reference as an in-line citation in the methods section.

Original Reviewer Comment: Line 94: Was severity classified for the entire 56.33 ha area that was burned? If so, what was the proportion that burned at high, moderate, and low severity? If not, how much of the burned area was assessed for severity?
Author Response: Because of the spatial heterogeneity of fire, it is difficult to accurately calculate the burning area of different severities. According to the field investigation and our classification criteria for different fire severities, the area of high: moderate: low severity is roughly 5:3:1. we added this information in the revised manuscript (line 91).
Reviewer Follow-up: Please provide the relative proportions of different severity levels as percentages instead of as a ratio. Also, please explain how you estimated the relative areas.

XXXX
XXXX


Original Reviewer Comment: Lines 94-95: Were the three plots in each burn severity area located in a single contiguous area of burn severity, or were they located in distinct burn severity patches that were separated by regions of variable burn severity?

Author Response: Every three plots were located in distinct burn severity patches.
The process of plots selection is as follows: field investigation of fired area→the fired area was divided into different parts (high, moderate and low severity areas) combined with the criterion→in every different part, we set three plots with similar slope, aspect and elevation.

Reviewer Follow-up: It is still unclear to me how the sample plots were distributed throughout the severity matrix. I.e., are the three high severity sample plots located in three distinct high severity patches? Or were the three high severity sample plots located in one single high severity patch? Please clarify this in the manuscript text. Adding more detail to figure S1 to show the entirety of the study area and the location of sample plots would also be of immense help.

XXXX
XXXX


Original Reviewer Comment: Line 160-162: Is it correct that two separate ANOVAs were conducted, one for each soil layer? Please clarify.
Response: I think it is appropriate to apply two separate ANOVAs in different soil layers, respectively. Because the topsoil and subsoil were considered as two different parts, not a whole in this research.
Reviewer Follow-up: Revise the text to clarify that you performed separate ANOVAs for each soil layer in the manuscript text. It currently remains unclear in the text.

XXXX
XXXX


Original Reviewer Comment: Lines 190-195: This speculation about the reasons for differences in SM belongs in the discussion section, not the results.

Author Response: Actually, we have made a discussion about the soil properties after fire in a paper we have published (Li et al., 2019). So we don’t make a special discussion about the soil and vegetation characteristics in the discussion part. We've only made a brief discussion of the interesting phenomena in this part of the results. Besides, we haven’t found a suitable place in the discussion part to put this content. We consider that this part of the content in the results does not affect the overall structure of the article. If not necessary, we want to leave this part of the content here.

Reviewer Follow-up: This belongs in the discussion. If necessary, revise the discussion to find a suitable place to include this information.

XXXX
XXXX


Original Reviewer Comment: Lines 254-263: Please report r2 values and p-values for each correlation referenced in this paragraph.

Author Response: Thank you for your advice. However, the r values for each correlation had been shown in Table S5 & Table S6 and we also used clear identifications (“*” , “**” ) to point out the correlation on significant levels. So we don’t think it’s necessary to report r2 values and p-values for each correlation in manuscript again.

Reviewer Follow-up: Include r2 values and p-values in this paragraph as well so readers don’t have to navigate to a supplementary table to assess the strength of the relationships. This is an important component of your results and should be included in the main text.

XXXX
XXXX


Original Reviewer Comment: Lines 296-297: Please explain how and why Acidobacteria promote nutrient cycling, plant growth, and soil restoration after disturbances. How does this differ from other bacteria phyla? Do different Acidobacteria subgroups play different roles?

Author Response: We rewrote this sentence to reduce ambiguity and the view of this sentence could be supported by the cited reference. Please see lines 299-301 in the revised manuscript.
Among them, Acidobacteria were found to play a role in promoting soil carbon cycling, as it can degrade complex plant derived polysaccharides such as cellulose and lignin (Huang et al., 2015).

Reviewer Follow-up: The sentence you provide in your response does not match the sentence in the manuscript. Revise manuscript to match the sentence provided in your response.

XXXX
XXXX


Original Reviewer Comment: Lines 346-347: Please specify how these factors are beneficial for bacterial communities and how they might impact the differences you observed.

Author Response: Actually, we have made a brief explanation that why the change of canopy density will be beneficial for bacterial communities.
Further, changes in plant community structure, such as the decline in canopy density, increase the light availability and soil temperature in the forest, which are beneficial for the bacterial communities (Arunachalam & Arunachalam, 2000; Bolat, 2014).

Reviewer Follow-up: My original comment was not related to canopy density, rather it was regarding lines 346-347, where you mention climate change, soil metals, and soil enzyme activities. How are these factors relevant to your study? Revise manuscript accordingly.

XXXX
XXXX


Original Reviewer Comment: Figure 1 caption: Please also specify the forest type, time since fire, and define the abbreviations used in the x-axis.

Author Response: The information of forest type and buning time have been declared in the M&M part (lines 82, 89).
We have made a supply in the figure 2 legend to define the abbreviations used in the x-axis.

Reviewer Follow-up: Please also specify forest type and time since fire in the figure caption, in addition to providing it in M&M lines 82 and 89.

XXXX
XXXX


Original Reviewer Comment: Table S1: Please spell out the fire severity level rather than using single-letter abbreviations. Cite your source in the caption to clarify where this severity classification system came from (it looks similar to one by Ryan & Noste, but if you modified a published severity classification system, then please provide a clear rationale for how & why it was modified. If you used a published severity classification system, then just report the original classification descriptions for each category…the current descriptions in this table are very unclear. Please revise the criteria description for the moderate severity category to be more descriptive. Use the term “low-severity,” not “low-grade.” Revise “height” in the moderate severity description to read “high severity.” Capitalize “unburned” in the control description. Please include this table in the main manuscript rather than as supplementary information. What does “burned down” mean? Do you mean consumed? Charred but still present?

Author Response: The full names of fire severity levels have been shown in the table.
The relevant reference has been added now.
We made a revision of the moderate severity and replaced the inappropriate words.
“unburned” was capitalized in the control description.
We have adjusted this table to the main manuscript.
“burned down” means consumed.

Reviewer Follow-Up: Revise “burned down” to read “consumed.”

XXXX
XXXX


Original Reviewer Comment: Figure S1: Add a scale bar. Also please add another panel to this figure to show the location and distribution of your research plots within your study site.

Author Response: The scale bar was added in fig. S1. We don't have more accurate remote sensing data. So more precise sample locations can not been shown in this figure.

Reviewer Follow-Up: The scale bar still does not appear in fig S1. Remote sensing data is not necessary to add more detail regarding the research area. Please provide an additional panel that shows the specific research area “zoomed in” to finer scale. Within this additional panel, add points to the map that show the location of the sample plots within the research area using either GPS or map coordinates. Use different symbology for the map points based on the severity level of each sample plot.

---

## Round 0.3 · accepted · Accept

The context of the manuscript within the larger research field on acidobacteria is much clearer now. Nice job. I have made a few corrections concerning spelling, formatting, etc., and added a sentence to clarify one of Reviewer #2 questions about the location of field plots (lines 106-107 in attached PDF). All edits are highlighted in yellow in the PDF document.

Finally, the fire took place in April 2015 according to text in the Materials and Methods Section, but in Figure 1, it is reported that the pictures were taken in October 2010 (before the fire)? Please address the discrepancy in the dates before publishing the manuscript in its final form.